# Diversity, Seasonal and Diel Distribution Patterns of Anchovies (Osteichthyes) in a Protected Tropical Lagoon in the Southwestern Gulf of Mexico

Guadalupe Morgado-Dueñas [1,2] and Manuel Castillo-Rivera [1,*]

1   Laboratorio de Peces, Departamento de Biología, Universidad Autónoma Metropolitana, Unidad Iztapalapa, Mexico City 09310, Mexico
2   PhD Program Doctorado en Ciencias Biológicas y de la Salud, Universidad Autónoma Metropolitana, Mexico City 09310, Mexico
*   Correspondence: crma@xanum.uam.mx

**Abstract:** Anchovies are species of ecological and economic importance that inhabit coastal waters, where they are very abundant. The objective of the study was, through high-frequency temporal sampling, to analyze the diversity, seasonal and diel distribution patterns of anchovies and their relationships with environmental variables. For 19 months, 24-h monthly cycles were carried out, taking samples every two hours. Permutational analysis of variance (PERMANOVA) and redundancy analysis (RDA) were used for data analysis. Seven species were captured, of which, *Anchoa mitchilli*, *Anchoa hepsetus*, *Anchoa lyolepis*, *Anchoa lamprotaenia* and *Cetengraulis edentulus* are common in brackish waters; however, *Anchoviella perfasciata* and *Engraulis eurystole* rarely occur in these systems. For these species, no major threats are known; therefore, they are listed as 'Least Concern'. A seasonal succession shows pulses during the closed-mouth phase and during the late warm-rainy season. At diel level, *A. mitchilli*, *C. edentulus* and *A. lamprotaenia* showed a markedly nocturnal pattern. RDA correlations showed that salinity, day/night effect, inlet state and rainfall were the most important factors related to anchovy distribution. Segregation along a salinity gradient was observed, in which *A. mitchilli* was captured mainly at low salinities, while other species occurred mainly at intermediate to high salinities. High-frequency sampling allowed a better understanding of the species richness and abundance patterns of anchovies in the system.

**Keywords:** environmental effects; food availability; new records; Ramsar; species diversity; temporal distribution

## 1. Introduction

Anchovies (Osteichthyes: Engraulidae) include 17 genera with approximately 143 species [1], of which 10 species in four genera occur in the Gulf of Mexico [2,3]; although only eight of them have been recorded on the continental shelf of Veracruz State in Mexico [4]. Most species are marine, although some enter to estuarine and freshwater systems, occurring in tropical and temperate coastal waters. Adults feed on small planktonic and bottom-living animals (a few by filter-feeding); however, large individuals may consume small fish. Although usually small (commonly 60 to 110 mm total length), many species school in such numbers that they form the basis of sizeable fisheries. Anchovies spawn in estuaries and sounds, as well as on the inner continental shelf. In the southern Gulf of Mexico, spawning regularly takes place during spring and summer or winter, while the minimum size at maturity for females is estimated to be between 37 mm and 97 mm [1,2].

Anchovies are a very important group ecologically and economically, since many species are commercially exploited for food or for the production of other products for human consumption; even in some countries, they are one of the main sources of economic

income in fishing activities [5]. As consumers of plankton, they are the link between primary production and higher trophic levels, playing an important role as a critical link in estuarine and coastal oceanic food webs [6–8]. In this way, they can control productivity in estuaries and their biomass level can limit fish production [9]. Several species of this family are frequently among the most abundant fish in coastal areas, both in juvenile and adult stages [10–13].

Estuarine fish populations show strong seasonal and diel variations, which are determined by abiotic factors, such as salinity, temperature, dissolved oxygen and rainfall [12,14,15]; and by biotic factors, such as recruitment patterns, competition, predation and food availability [11,16,17], which can act separately or in synergy. In addition, in many studies of estuarine communities, it has been observed that during the twilight or at night, a greater number of fish species is caught [12,16,18,19]. For this reason, samplings throughout 24 h cycles can provide a better understanding of fish richness and abundance [11,12].

In this sense, describing and analyzing the seasonal and diel variation in the abundance of anchovies can contribute to a better understanding of their diversity and improve the management of this resource. Although there are several studies on the distribution and abundance of anchovies at community and population levels, particularly for some species such as *Anchoa mitchilli* and *Cetengraulis edentulus* [20–22], there are fewer studies on other anchovies in the western central Atlantic; in particular, those that analyze 24 h cycles. In the Southwest Atlantic, anchovies show differences in their behavior on a temporal scale, both seasonally and in 24 h cycles, due to the influence of environmental factors, since these regulate various processes, such as spawning, feeding, reproduction and recruitment. Therefore, their populations are subject to large fluctuations caused by environmental variability [10,23].

Under the hypothesis that definite changes will be observed in the distribution of anchovies at the seasonal level, mainly as a function of variations in salinity, temperature, precipitation and inlet stage, and at the diel level, primarily associated with the light/dark period, the aim of this study was to determine the species richness of anchovies, and to analyze the seasonal and diel patterns of the distribution of these species, as well as the influence exerted by environmental factors on these patterns in La Mancha lagoon.

## 2. Materials and Methods

### 2.1. Study Area

La Mancha lagoon is a Ramsar site, a category defined as a site that provides for national action and international cooperation regarding the conservation of wetlands and wise sustainable use of their resources. This system is located in the Veracruz state, Mexico (19°33′55′′–19°35′44′′ N and 96°22′41′′–96°23′39′′ W). It is a small intermittent microtidal system, with an approximate extension of 1.742 km² [24,25] (Figure 1) and surrounded by a mangrove forest. At its northeastern end, this lagoon is connected to the sea via an inlet that discharges through a sand barrier, which during study period was closed from January to May. The region has a warm, subhumid climate (Aw2), with two climatic seasons: a dry season (mean monthly rainfall < 60 mm) from November to May and a rainy season (mean monthly rainfall > 100 mm) from June to October.

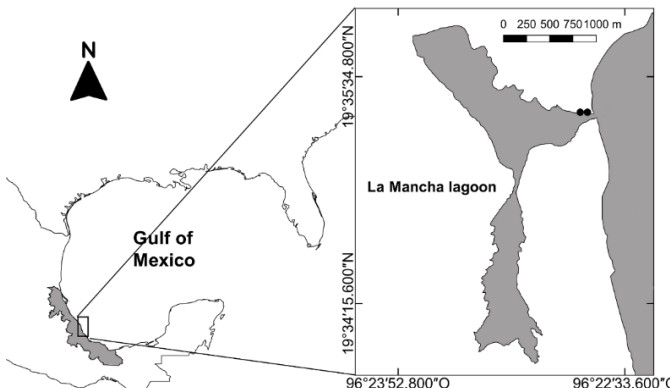

**Figure 1.** Geographical location of La Mancha lagoon and sampling points (black circles).

### 2.2. Data Collection

In a sandy habitat inside the lagoon, during 19 months from April 2012 to October 2013, every month 24-h cycles were carried out simultaneously at two sites (one at 325 m and the other at 225 m from the inlet system), taking samples every two hours (a total of 442 samples). Samples were taken during the new moon to maximize light/darkness and tidal effects, using a beach seine net (37 m long, 1.2 m deep and 1 cm mesh size) in water up to about 1.4 m deep, covering an approximate sampling area of 1500 m². Immediately after capture, fish were anesthetized (clove oil solution), fixed with 10% formalin and preserved with 70% ethanol. Individuals were identified according to specialized literature [2,26] and counted in the laboratory. The main taxonomic characters to discriminate the species were: number of gill rakers on first arch; lengths of maxilla, snout, axillary scale and pseudobranch; position of anal-fin origin and anus; extension of first pectoral fin ray; and number of rays in the anal fin.

Simultaneously in each sampling, sampling time, salinity, water temperature, dissolved oxygen, chlorophyll *a* and zooplankton biomass were recorded in situ. Water level (tidal stage effect) for each sampling time was recorded from regional data for the state of Veracruz. To evaluate the influence of long-term data with a regional incidence of important environmental conditions, monthly averages (corresponding to 60 years) of precipitation and atmospheric temperature were also considered [27]. These long-term regional variables can exert an important influence on fish distribution as those recorded in situ [28].

### 2.3. Data Analysis

For the most frequent species (>10% of frequency of capture), two-way univariate permutational analysis of variance (PERMANOVA) was applied to evaluate the significance of the differences in the number of fish among months and between day and night, as well as their interactions. PERMANOVA is a routine for testing the simultaneous response of one or more variables to one or more factors, on the basis of a resemblance measure, generally highly appropriate, because most ecological data (being counts of abundances of species) tend to be overdispersed, with a substantial proportion of zeros. This method is also robust because it uses multiple random permutations to obtain *p*-values; thus, normality and homogeneity of variances are directly implied by the permutation procedure [29,30]. PERMANOVA was performed using Euclidean distance (on square root transformed data) and permutated residuals under a reduced model, Type III (maximum permutations = 999), according to the routine for univariate analysis [30]. This analysis was performed using the software PRIMER v7 [31].

Redundancy analysis (RDA) was applied to the abundance data matrix of all the species (dependent set) and to the environmental data matrix (independent set), in order to elucidate the relationships between these two data groups. The significance of environmental factors was determined using 499 unrestricted Monte Carlo permutations. Correlations between environmental variables and species-derived sample scores were

used to determine the relationship between these data sets [32]. Ordination analyses were performed using the package CANOCO ver. 4.5.

## 3. Results

A total of 8,465 individuals of *Anchoa mitchilli* (Valenciennes, 1848), 1415 of *A. hepsetus* (Linnaeus, 1758), 1,003 of *A. lamprotaenia* (Hildebrand, 1943), 150 of *A. lyolepis* (Evermann & Marsh, 1900), 343 of *Cetengraulis edentulus* (Cuvier, 1829), 4 of *Anchoviella perfasciata* (Poey, 1869) and 1 of *Engraulis eurystole* (Swain & Meek, 1884) were caught.

The monthly variation of the most abundant species showed significant pulses throughout the 19 studied months (Table 1). Higher mean values of *A. mitchilli* were in May and October of the first year, while higher pulses of *A. hepsetus* were observed in April, also during the first year. For *C. edentulus*, two pulses were observed, the first and greater in February and the second during September–October of the first year. By contrast, *A. lamprotaenia* shows pulses in September–October, but of the second year (Figure 2).

**Table 1.** Results from univariate two-way permutational analysis of variance (PERMANOVA) to test effects of month and diel period (day/night) on the relative abundance of species.

| Source of Variance | df | MS | Pseudo-F | P (Permuted) |
|---|---|---|---|---|
| *Anchoa mitchilli* | | | | |
| Months | 18 | 14,046 | 8.405 | 0.0001 |
| Diel | 1 | 39,377 | 23.563 | 0.0001 |
| Months x Diel | 18 | 1360.2 | −0.81396 | 0.0959 |
| Residual | 404 | 1671.1 | | |
| *Anchoa hepsetus* | | | | |
| Months | 18 | 1220.3 | 6.8875 | 0.0001 |
| Diel | 1 | 208.57 | 11772 | 0.2104 |
| Months x Diel | 18 | 219.08 | −1.2365 | 0.2347 |
| Residual | 404 | 177.18 | | |
| *Anchoa lamprotaenia* | | | | |
| Months | 18 | 336.83 | 5.8138 | 0.0001 |
| Diel | 1 | 878.04 | 15.155 | 0.0001 |
| Months x Diel | 18 | 64.768 | −1.1179 | 0.2749 |
| Residual | 404 | 57.937 | | |
| *Cetengraulis edentulus* | | | | |
| Months | 18 | 27.303 | 4.4635 | 0.0001 |
| Diel | 1 | 87.311 | 14.274 | 0.0005 |
| Months x Diel | 18 | 5.1598 | 0.84353 | 0.0222 |
| Residual | 404 | 6.1169 | | |

At diel level, *A. mitchilli*, *A. lamprotaenia* and *C. edentulus* showed significant differences between day and night (Table 1), with higher values at night (20:00 to 04:00 h), decreasing at sunrise (06:00 h); lower values in the day (08:00 to 16:00) and increasing again at dusk (18:00 h) (Figure 3). On the other hand, *A. hepsetus* did not show significant differences in abundance at diel level. Interaction between seasonal and diel factors was only significant for *C. edentulus*, because this species in January was more abundant during daylight hours (Table 1).

The RDA results showed that first two ordination axes explained 98.2% of the cumulative constrained variance in the species–environment relations. According to significance tests derived from this analysis, salinity, diel effect, atmospheric temperature, inlet state, mean precipitation and water temperature showed a significant relationship with species distribution. Correlations between these environmental variables and species-derived sample scores were relatively high, except for water temperature (Table 2). Variance inflation factors (VIF) also showed that the measurements are not strongly correlated.

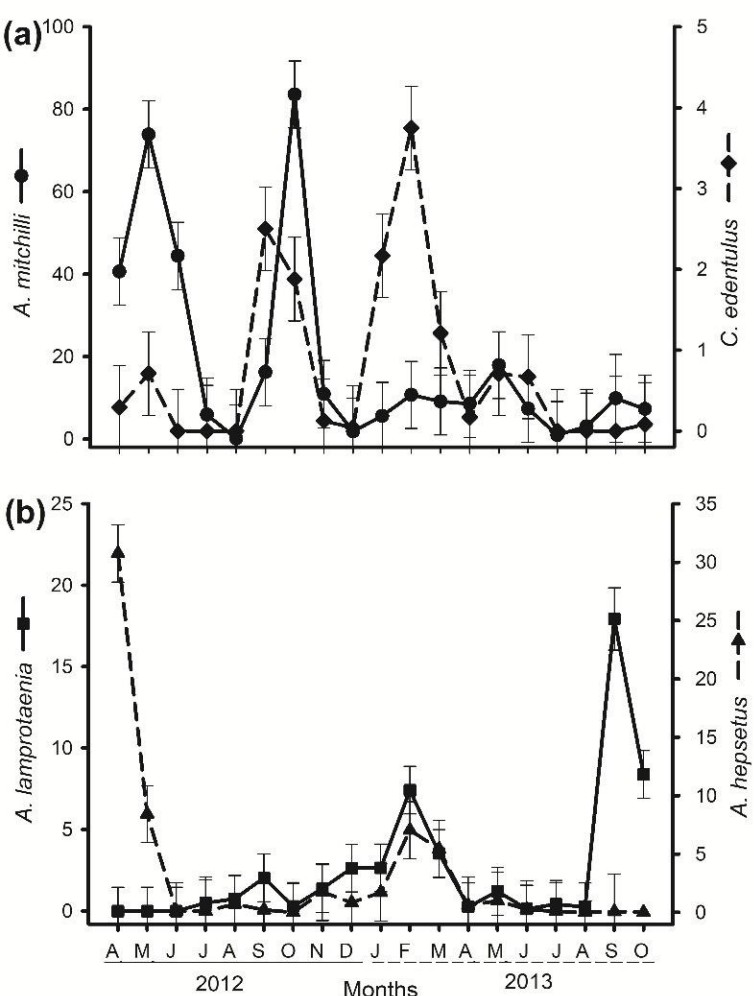

**Figure 2.** Monthly mean number and standard error for (**a**) *A. mitchilli* and *C. edentulous*; and (**b**) *A. hepsetus* and *A. lamprotaenia*.

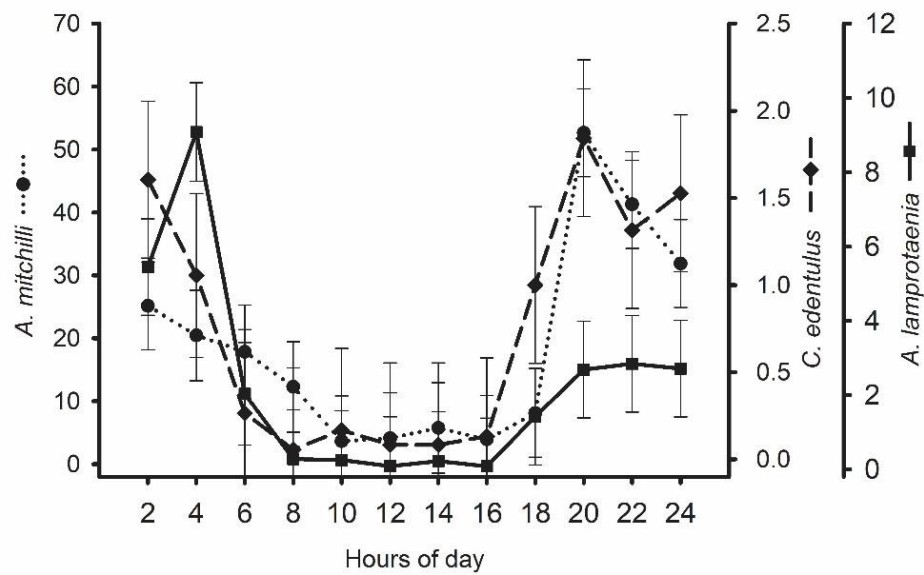

**Figure 3.** Mean and standard error values by hour of *A. mitchilli, A. lamprotaenia* and C. *edentulus*.

**Table 2.** Correlations between environmental variables and species-derived sample scores, significance and variance inflation factors (VIF) from redundancy analysis (RDA).

| Correlations | Axis 1 | Axis 2 | P | VIF |
|---|---|---|---|---|
| Salinity | −0.300 | 0.0006 | 0.002 | 1.911 |
| Diel effect | −0.286 | 0.019 | 0.002 | 1.035 |
| Open inlet | −0.172 | 0.221 | 0.026 | 2.707 |
| Atmospheric temperature | 0.119 | 0.127 | 0.002 | 2.143 |
| Rainfall | −0.063 | 0.162 | 0.014 | 3.142 |
| Water temperature | 0.032 | 0.043 | 0.030 | 2.395 |
| Zooplankton | −0.097 | 0.044 | 0.392 | 1.257 |
| Chlorophyll *a* | −0.027 | 0.053 | 0.258 | 1.288 |
| Dissolved oxygen | 0.116 | −0.041 | 0.162 | 2.368 |
| Water level | −0.039 | −0.070 | 0.750 | 1.072 |

These results reveal a main ordination gradient (Axis 1) mainly related to salinity and light-dark gradients, and in the opposite direction to atmospheric temperature, while another major trend (Axis 2) was related to inlet state and rainfall (Figure 4). As inlet state and diel period are binary categorical variables, the corresponding vector indicates, in the direction of the arrows, a preference for the open inlet and day period, respectively. In this sense, *A. mitchilli* and *C. edentulus* were mainly associated with warm conditions, intermediate to low salinities and at night, while *A. hepsetus* was related with the closed-mouth phase, intermediate salinities and low rainfall. Similarly, *A. lamprotaenia* was associated with intermediate to high salinities, low rainfall and at night. Thus, along a salinity gradient, *A. mitchilli* segregates from other anchovy species (Figure 5). Likewise, *A. mitchilli*, *A. hepsetus* and *C. edentulus* were captured mainly at night (Figure 3) and when the inlet was closed (January to May). The species *A. lyolepis*, *A. perfasciata* and *E. eurystole* were recorded at intermediate conditions of the environmental variables analyzed.

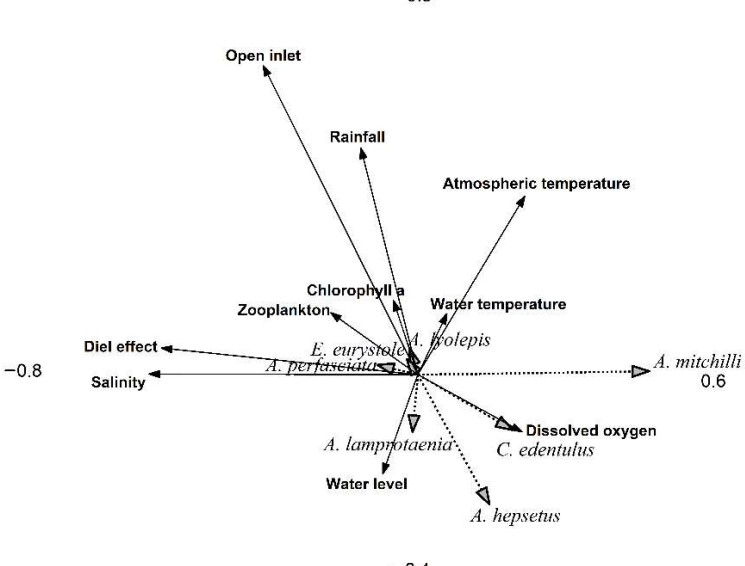

**Figure 4.** RDA biplot of species (dotted line vectors) and environmental variables (solid line vectors). Length and direction of arrows indicate the relative importance and direction of change of environmental variables.

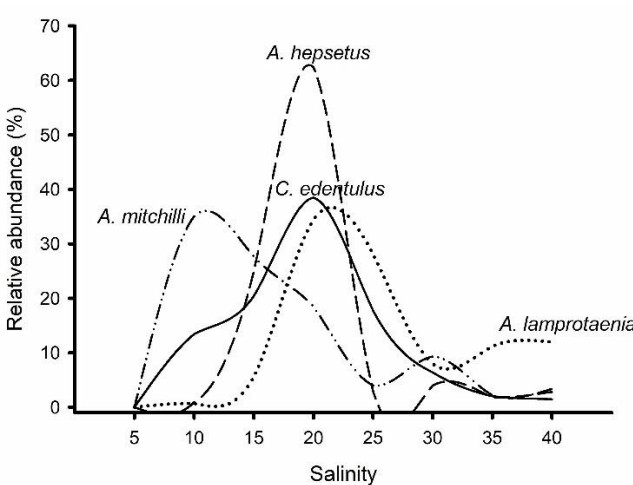

**Figure 5.** Percentage of captured individuals of each species, in relation to their own total abundance, according to a salinity gradient.

## 4. Discussion

Seven species were captured, which represents the highest number of anchovy species recorded for all estuarine environments in the southwestern Gulf of Mexico [4], mainly due in the intensive sampling used, both seasonal and diel. Five of these species, *A. mitchilli*, *A. hepsetus, A. lamprotaenia, A. lyolepis* and *C. edentulus*, are commonly found in brackish waters. In contrast, *A. perfasciata* and *E. eurystole* are marine species that rarely occur in lagoons and estuaries. Indeed, the capture of *A. perfasciata* represents a new record for Mexican estuaries.

Throughout the 19 months sampled, the most abundant species showed a seasonal succession, in which abundance pulses were observed during the closed-mouth phase (January–May) and during the late warm-rainy season (September–October). Thus, *A. mitchilli* and *C. edentulus* showed higher capture pulses in both periods of the first year. The major pulse of *A. hepsetus* was in the closed-mouth phase of the first year, while *A. lamprotaenia* was the only species with a higher pulse during the late warm-rainy season of the second year. This seasonal succession may be related to differences in physiological tolerances and in life cycle, favoring a partitioning resource.

Regarding diel variability, the simple daily periodicity of sunrise and sunset affects the activity of fish, which generates patterns of diel migrations between adjacent habitats usually related to feeding, shelter or reducing the risk of predation; avoidance of inter- and/or intraspecific competition; reproduction; and searching for a physiologically optimum environment. Thus, species may become typically diurnal, nocturnal or crepuscular [12,33]. In La Mancha lagoon, *A. mitchilli, C. edentulus* and *A. lamprotaenia* showed a markedly nocturnal pattern. The RDA results also showed that the day/night effect is one of the most important environmental factors in determining anchovy distribution patterns. This nocturnal capture pattern is very consistent for *A. mitchilli*, which has been observed in both temperate [11,34,35] and tropical coastal systems [12,18]. Although many studies have invoked that the diel movements of fish seem to be more related to feeding activity, food availability does not appear to be the answer, since it has been observed that *A. mitchilli* consume significantly less prey at night than during the day, being more abundant at night [35]. Thus, the greater capture of anchovies at night in shallow waters could be more related to shelter or strategies to avoid predation.

Although a high density of anchovies was recorded in a sandy habitat in the present study, it is difficult to assess the role that adjacent habitats, such as those dominated by mangroves, may play in this pattern, because it may be due to a preference for bare habitats or simply reflect migration processes to adjacent habitats.

RDA ordination explained a high percentage of the variance of species–environment relation (~99%), which implies that the environmental variables considered give an adequate

description of the ecological processes. Correlations from RDA also revealed that variables in situ (salinity, day/night effect and stage inlet) and with regional incidence (rainfall and atmospheric temperature) were the main factors associated to anchovy distribution.

As has been commonly observed, salinity has important direct effects on the distribution of fish in estuarine areas [13–15], as well as indirectly through the modification of the result of biotic interactions, such as competition between species [15,28,36]. Although the most abundant species recorded are able to tolerate a wide range of salinities, entering the brackish waters of lagoons and estuaries, segregation along a salinity gradient was observed in La Mancha lagoon, in which *A. mitchilli* was captured primarily at low salinities, while *A. hepsetus, A. lamprotaenia* and *C. edentulus* occurred mainly at intermediate to high salinities. Indeed, *A. mitchilli* is a euryhaline species regularly caught at low salinities, as has been observed in temperate and tropical estuaries [12,37,38]. The separation of anchovy species related to salinity variation has also been observed in tropical systems of Brazil [10,23]. This segregation pattern is associated to the physiological tolerances of anchovies and may also be related to their competitive capabilities, since it is relatively common for eurytopic species to be weaker competitors, being displaced by stenotopic species, which tend to be stronger competitors. Both processes can allow a differential use of resources.

Many studies in intermittently open estuaries have shown that fish abundance can undergo significant seasonal changes, mainly related to changes in estuarine mouth phase and salinity regime [39–42]. Although in these systems the immigration of marine species is limited during the closed-mouth phase, some estuarine species are more abundant in this period [39–41]. In La Mancha lagoon, pulses of abundance of *A. mitchilli, A. hepsetus* and *C. edentulus* were observed when the mouth is closed, as the fish were retained within the estuary, probably taking advantage of more stable physical conditions, high water levels and increased food availability [40]. Indeed, significant pulses of numerical abundance of copepods have been observed in this system during February–March [43].

Rainfall plays an important role in determining the seasonal patterns of salinity and mouth opening regimes. In general, higher rainfall in tropical latitudes also increases river discharge and riverine freshwater runoff, and brings an increased amount of allochthonous organic matter and dissolved nutrients to systems, leading to higher food availability at the end of the rainy period. In this sense, the importance of rainfall on the abundance of estuarine fish, mainly in tropical and subtropical systems, has been observed in many studies [13,14,44,45]. In La Mancha lagoon, pulses of abundance of anchovy species were observed during the late warm-rainy season, which coincide with pulses of numerical abundance of decapod larvae, recorded for the system during September–October [43]. Thus, an increase in food resources would favor fish immigration to the estuarine system for food [17,18].

RDA results showed that two factors with long-term regional incidence, such as rainfall and atmospheric temperature regimes, can be important drivers in determining distribution patterns of anchovies. In this sense, these factors can have an incidence as important as the variables recorded in situ; thus, the environmental processes operating at local and regional scales may act synergistically on the distribution patterns of species at the local scale. In contrast, correlations between environmental factors and species-derived sample scores also revealed a low effect on anchovy distribution of some environmental variables recorded in situ, such as water temperature, dissolved oxygen and chlorophyll *a*. Indeed, atmospheric temperature had a greater influence on anchovy distribution than water temperature. Thus, long-term regional variables would be reflecting adaptive responses of fish.

Zooplankton density (mainly copepods) has been shown to be an important factor determining anchovy abundances [8,46,47]; however, RDA results also show a non-significant effect of zooplankton biomass. Thus, although the anchovy pulses coincide with the numerical pulses of the abundance of copepods (February–March) and decapod larvae (September–October) reported for the system, the zooplankton biomass employed does

not consider these groups separately. Likewise, the tidal state had little influence mainly because the southwestern coastline of the Gulf of Mexico is predominantly microtidal [48].

With the exception of *A. mitchilli*, there is little information on the distribution and abundance of these anchovy species in the southwestern Gulf of Mexico, particularly those that analyze 24-h cycles. According to IUCN [49], only *A. mitchilli* has a current population trend of stable; however, this status is unknown for the other species due to the limited information available. Furthermore, for these species, there are no known major threats; therefore, they are listed as 'Least Concern'.

Although they are currently species only harvested by artisanal fishermen in the study area, as in other countries, they represent a potential resource. In this way, the information provided on seasonal and diel variations in the abundance of anchovies in the present study can be useful for a better management of the resource. Likewise, this information can be important to know the diversity and improve the ecosystem management of a Ramsar protected wetland.

## 5. Conclusions

High-frequency temporal sampling provided a better understanding of patterns of species richness and the abundance of anchovies in a protected tropical coastal lagoon. Throughout the 19 months sampled, a seasonal succession of species was observed, with capture pulses during the closed-mouth phase, as species were retained within the system and during the late warm-rainy season, when the main migration of fish into the lagoon occurs, taking advantage of favorable conditions. In this sense, anchovy pulses coincide with numerical pulses of the abundance of copepods and decapod larvae reported for the system; although the RDA did not show a significant correlation between the abundance of anchovies and zooplankton biomass. At diel level, three of the most abundant species (*A. mitchilli, C. edentulus* and *A. lamprotaenia*) showed a consistent nocturnal pattern, which may be related to shelter or avoidance of predation. According to RDA ordination, salinity, day/night effect, inlet state, atmospheric temperature and rainfall were the main drivers in determining anchovy distribution patterns. Thus, environmental factors with local (in situ) and regional incidence can act synergistically on the distribution patterns of anchovies in La Mancha lagoon. Capture pulses may be related to differences in physiological tolerances. In this way, it was observed that along a salinity gradient, the most abundant species (*A. mitchilli*) was mainly capture at low salinities, while *A. hepsetus, A. lamprotaenia* and *C. edentulus* at intermediate to high salinities. This segregation may also be mediated by competitive processes.

**Author Contributions:** Conceptualization, M.C.-R. and G.M.-D.; formal analysis, M.C.-R. and G.M.-D.; data curation, G.M.-D. and M.C.-R.; writing—original draft preparation, M.C.-R. and G.M.-D.; writing—review and editing, M.C.-R. and G.M.-D.; visualization, M.C.-R. and G.M.-D. All authors have read and agreed to the published version of the manuscript.

**Funding:** This research was funded by the Universidad Autónoma Metropolitana, Unidad Iztapalapa: 14302047. Guadalupe Morgado-Dueñas had the support of the Consejo Nacional de Ciencia y Tecnología (CONACyT), through the scholarship: CVU– 868711.

**Institutional Review Board Statement:** All the biological collections were authorized and approved by the Cooperativa Pesquera La Mancha (fishermen's cooperative). All the studied species are not endangered or protected in Mexico; thus, it was not necessary to apply for any other licence [Norma Oficial Mexicana NOM-059-SEMARNAT-2010].

**Informed Consent Statement:** Not applicable.

**Data Availability Statement:** Not applicable.

**Acknowledgments:** We thank the academic editors and the constructive comments on the manuscript from three anonymous reviewers were very valuable. We also thank Juan José Ambriz for his important support.

**Conflicts of Interest:** The authors declare no conflict of interest.

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
