# Peer review of "Diversity, Seasonal and Diel Distribution Patterns of Anchovies (Osteichthyes) in a Protected Tropical Lagoon in the Southwestern Gulf of Mexico"

_diversity, doi:10.3390/d14100852_

Round 1
Reviewer 1 Report
First of all, I am really happy and enjoy to read the manuscript. The research is well done with the intensive survey for 19 months with 8465 anchovies. Recently, I do not read such an intensive research papers in the relevant research. Hence, the manuscript definitely fits to the journal. The authors examined not only the species occurrence of anchovies but also the integration of the occurrences with environmental factors. The statistical analyses also supports the environmental factors affecting the occurrence of each species in the study area. The target species is also important for fisheries and the study site is also an important as a Ramsar site. This manuscript will be useful to fisheries management and conservation in the future. I have some minor suggestions only as follows.
1, I did not find the study carried out in what year. Please state in text and figure caption in Fig. 2.
2, L95: It would be better to state the identification methods in short not only refer the previous works.
3, Fig. 2: Abbreviation of each month would be better as one word. M, A, M, J, J, A… as the study has been monthly. Please add years.
4, Fig. 3: the error bars should be thicker like Fig. 2.
5, It would be better to state/discuss the current stock status in each species in abstract/discussion as anchovies are highly commercial species. Are all species healthy in terms of biodiversity? Or, some are vulnerable or other statuses etc.
Reviewer 2 Report
The present study aims to determine the diversity, in terms of number of species, of anchovies, and to analyze the seasonal and diel patterns of distribution of these species, as well as the influence exerted by environmental factors on these patterns in La Mancha lagoon.
Overall, the manuscript is clear and well structured. It is fairly easy to read and to interpret the main results. The methods were properly applied and the statistical analysis adequate to answer the maing goals of the work.
The article’s main drawback is the lack of novelty. Although the 24 hour sampling effort is not frequently done, there are not new topics to explore and no new insights are added to the existing ones. Nonetheless, I believe the effort done by the authors to conduct the sampling and the obtained results are worth to be published in Diversity, after some revisions are considered by the authors, namely the following:
- Introduction: the authors present the importance of the species considered, namely regarding their ecological and economic importance, with many species being commercially exploited. This appears as a fundamental factor for authors to have chosen these species and the considered sampling effort, which will allow to analyze the seasonal and diel variation in the distribution and abundance of anchovies, which in turn will contribute to a better understanding of their diversity and improve the management of this resource. All of this makes sense, but then it seems that something is missing: in one hand, there is no information about the exploration status of these species. Are they overexploited or exposed to other type of impact that threatens their presence in the Lagoon? On the other hand, it seems to be missing the link for this topic, which is presented as one of the main reasons for this species to be studied, with the results of the work. Thus, I was expecting to read something in the discussion regarding this topic, namely how the results could influence and contribute for the appropriate management of this resource, including their diel and seasonal variation. I believe that if the authors are able to add this in the discussion, it will significantly improve the work, namely its usefulness for practical purposes and applicability.
- regarding the study area, it is described as surronded by mangrove forest. As the environmental factors included in the analysis explained a high percentage of the species variability, is it expected that the role of this adjacente habitat to be residual or even not to play any role in the number of species of anchovies or their abundance and biomass?
- how many 24 hours sampling cycles were performed in total? Please add this information in the data collection section.
- Also regarding data collection, I understand that environmental data collected in-situ and regional data may be importante in explaining variation patterns of these species (either for number of species, densities or biomass), but I couldn’t understand why the regional variables were collected for 60 years? Were an average used for each factor? Why not using the values for that specific period? How can we understand and distinguish between the contribution of pontual measurements with long term data?
- Line 107 (in data analysis): “For the most frequente species (>10% of frequency of capture) two-way PERMANOVA was applied to evaluate the diferences in the number between months and day and night, as well as interactions”… number of what?
Lines 53- 55: reference is missing to state that a higher number of species is caught at night
Figure 5 – Plots with many axis may be difficult to read and interpret (although I understand that differences in number of individuals are huge between species which would be also difficult to represente graphically). Is there a possibility to calculate densities instead of number of individuals?
Lines 239-341: reference is missing.
Reviewer 3 Report
Good job, clearly presented and well analysed.
Suggestions : give some general informations about differences in the life cycles of the different species (if known) like reproductive and feeding habits, sizes and growth, fecundity ...). This will help the non specialist (as myself) to take advantages of your results
line 76 : define "Ramsar"
